# Experimental realization of two-dimensional Dirac nodal line fermions in monolayer Cu$_2$Si

Baojie Feng [1,2], Botao Fu[3], Shusuke Kasamatsu[1], Suguru Ito[1], Peng Cheng[4], Cheng-Cheng Liu [3], Ya Feng[2,5], Shilong Wu [2], Sanjoy K. Mahatha[6], Polina Sheverdyaeva [6], Paolo Moras [6], Masashi Arita[2], Osamu Sugino[1], Tai-Chang Chiang[7], Kenya Shimada[2], Koji Miyamoto[2], Taichi Okuda[2], Kehui Wu[4], Lan Chen[4], Yugui Yao[3] & Iwao Matsuda[1]

Topological nodal line semimetals, a novel quantum state of materials, possess topologically nontrivial valence and conduction bands that touch at a line near the Fermi level. The exotic band structure can lead to various novel properties, such as long-range Coulomb interaction and flat Landau levels. Recently, topological nodal lines have been observed in several bulk materials, such as PtSn$_4$, ZrSiS, TlTaSe$_2$ and PbTaSe$_2$. However, in two-dimensional materials, experimental research on nodal line fermions is still lacking. Here, we report the discovery of two-dimensional Dirac nodal line fermions in monolayer Cu$_2$Si based on combined theoretical calculations and angle-resolved photoemission spectroscopy measurements. The Dirac nodal lines in Cu$_2$Si form two concentric loops centred around the Γ point and are protected by mirror reflection symmetry. Our results establish Cu$_2$Si as a platform to study the novel physical properties in two-dimensional Dirac materials and provide opportunities to realize high-speed low-dissipation devices.

[1] The Institute for Solid State Physics, The University of Tokyo, Kashiwa, Chiba 277-8581, Japan. [2] Hiroshima Synchrotron Radiation Center, Hiroshima University, 2-313 Kagamiyama, Higashi-Hiroshima 739-0046, Japan. [3] Beijing Key Laboratory of Nanophotonics and Ultrafine Optoelectronic Systems, School of Physics, Beijing Institute of Technology, Beijing 100081, China. [4] Institute of Physics, Chinese Academy of Sciences, Beijing 100190, China. [5] Ningbo Institute of Materials Technology and Engineering, Chinese Academy of Sciences, Ningbo 315201, China. [6] Istituto di Struttura della Materia, Consiglio Nazionale delle Ricerche, I-34149 Trieste, Italy. [7] Department of Physics, University of Illinois, Urbana, IL 61801, USA. Baojie Feng and Botao Fu contributed equally to this work. Correspondence and requests for materials should be addressed to L.C. (email: lchen@iphy.ac.cn) or to Y.Y. (email: ygyao@bit.edu.cn) or to I.M. (email: imatsuda@issp.u-tokyo.ac.jp)

The discovery of topological insulators has ignited great research interest in the novel physical properties of topological materials in the past decade[1, 2]. A characteristic feature of topological insulators is the existence of topologically nontrivial surface states that are protected by time reversal symmetry. Recently, tremendous research interest has moved from traditional topological insulators to topological semimetals which have vanishing densities of states at the Fermi level. The valence and conduction bands in topological semimetals can touch at either discrete points or extended lines, forming Dirac/Weyl semimetals[3–6] and nodal line semimetals[7–18]. The band degeneracy points or lines in topological semimetals are also protected by symmetries and are thus robust against external perturbations.

On the other hand, two-dimensional materials have also attracted broad scientific interest because of their exotic properties and possible applications in high-speed nano-devices[19]. Therefore, it is natural to ask the following question: do three-dimensional topological semimetals have counterparts in two-dimensional materials? The realization of two-dimensional topological semimetals will provide new platforms for the design of novel quantum devices at the nanoscale. For Dirac semimetals, a two-dimensional counterpart is graphene, when spin orbit coupling (SOC) is neglected[4, 5]. Two-dimensional nodal line fermions are predicated to exist in monolayer hexagonal lattices[20], honeycomb-kagome lattices[21] and monolayer transition metal-group VI compounds[22]. The nodal lines in these materials are protected by (slide) mirror symmetry and require negligible SOC. However, the experimental realization of such structures in real materials is quite challenging; thus, it is necessary to search for new and realizable two-dimensional materials that host robust nodal lines.

In this work, we study monolayer Cu$_2$Si, which is composed of a honeycomb Cu lattice and a triangular Si lattice. In the free-standing form, all Si and Cu atoms are coplanar[23] and thus

mirror reflection symmetry with respect to the $xy$ plane ($M_z$) is naturally expected; this is important for the existence of two-dimensional nodal lines. Importantly, the experimental synthesis of monolayer Cu$_2$Si is easy and has already been realized decades ago. One method to synthesize Cu$_2$Si is the direct growth of copper on Si(111)[24–26]. However, monolayer Cu$_2$Si forms a quasiperiodic superstructure on Si(111) without precise long range periodicity. Alternatively, a commensurate $1 \times 1$ structure of monolayer Cu$_2$Si has been synthesized on a Cu (111) surface using chemical vapor deposition (CVD) methods[27, 28]. As the $1 \times 1$ lattice of Cu$_2$Si approximately matches the $(\sqrt{3} \times \sqrt{3})$R30° superlattice of Cu(111), large domains of ordered Cu$_2$Si can form on the Cu(111) surface. However, previous investigations of Cu$_2$Si have primarily concentrated on the structural and chemical properties, and a detailed investigation of its band structures is still lacking. Here, our comprehensive theoretical calculations show the existence of two Dirac nodal loops centred around the Γ point. The gapless nodal loops are protected by mirror reflection symmetry. These intriguing band structures have been directly observed by angle-resolved photoemission spectroscopy (ARPES) measurements of Cu$_2$Si/Cu(111). Both nodal loops survive in the Cu$_2$Si/Cu(111) system because of the weak substrate-overlayer interaction.

## Results

**First-principles calculations on freestanding Cu$_2$Si.** The band structures of freestanding Cu$_2$Si without SOC are shown in Fig. 1b. Within 2 eV of the Fermi level, there are three bands: two hole-like bands (labelled α and β) and one electron-like band (labelled γ). All three bands form closed contours on the Fermi surface: a hexagon, a hexagram, and a circle, respectively, as shown in Fig. 1e. Interestingly, we find that band γ crosses bands α and β linearly in all directions without opening of energy gaps, thus forming two concentric Dirac nodal loops centred around

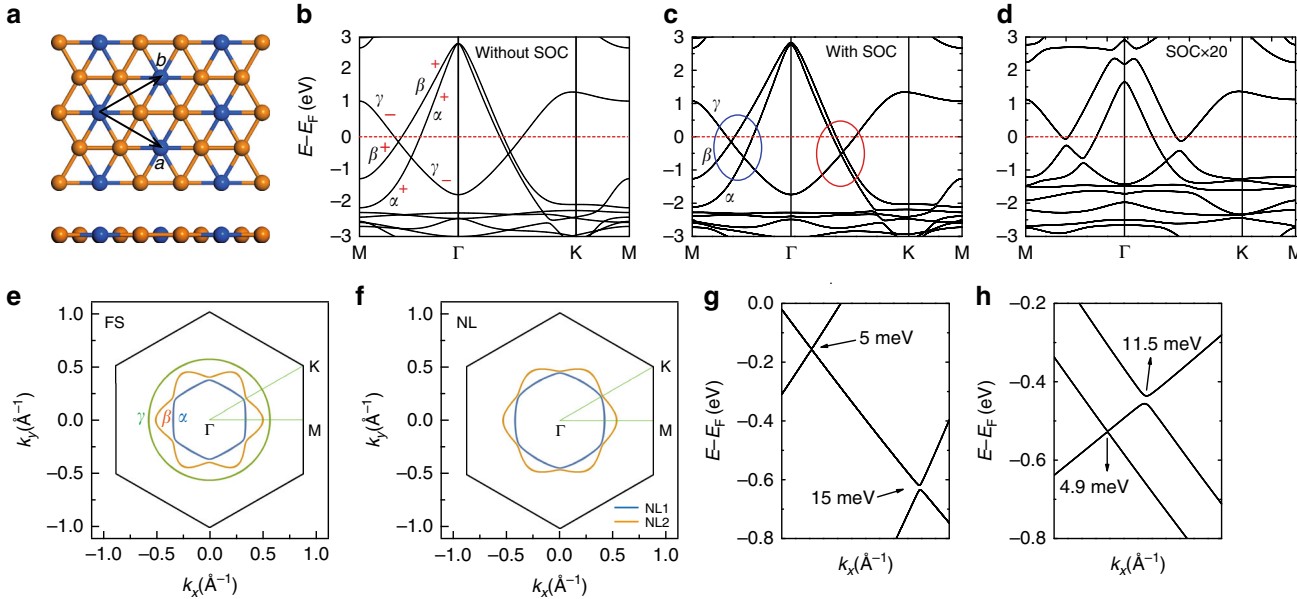

**Fig. 1** Atomic and band structures of free-standing Cu$_2$Si. **a** Top and side view. The orange and blue balls represent Cu and Si atoms, respectively. **b, c** Calculated band structures of Cu$_2$Si without and with spin-orbit coupling (SOC), respectively. The vertical axis $E-E_F$ corresponds to $-E_B$, where $E_B$ is the binding energy. For simplicity, we label the three bands that cross the Fermi level α, β, and γ, respectively. The parity of mirror reflection symmetry for each band is labelled plus and minus signs in **b**. The zoom-in band structures in the blue and red ellipses are shown in **g, h**. **d** Band structure of Cu$_2$Si after artificially increasing the intrinsic SOC by 20 times. **e** Fermi surface of Cu$_2$Si without SOC. The blue, orange, and green lines correspond to bands α, β, and γ, respectively. **f** Momentum distribution of the nodal loops: NL1 (blue) and NL2 (orange). **g, h** Zoom-in band structures in the blue and red ellipses in **c**, which clearly show the SOC-induced gaps

the Γ point (labelled NL1 and NL2). In Fig. 1f, we present the momentum distribution of gapless nodal points in the Brillouin zone, which shows a hexagon (NL1) and a hexagram (NL2), respectively.

It is natural to ask whether the two gapless Dirac nodal loops in $Cu_2Si$ are symmetry-protected. To answer this question, we calculated the $M_z$ parity of each band without SOC and find that the parity of band $\gamma$ is opposite to that of bands $\alpha$ and $\beta$, as indicated by the plus and minus signs in Fig. 1b (see Supplementary Note 1 for orbital analysis). The opposing $M_z$ parities indicate that band $\gamma$ does not couple with bands $\alpha$ and $\beta$, and therefore both Dirac nodal loops remain gapless. This result provides strong evidence that the two gapless nodal loops are protected by mirror reflection symmetry.

After including SOC, each band is double-degenerate and the two degenerate bands have opposing $M_z$ parity. As a result, the mirror reflection symmetry cannot protect the nodal loops anymore. Along the preceding Dirac nodal loops, the bands with the same parity will couple with each other, resulting in the opening of band gaps. The calculated band structures with SOC clearly show that the nodal lines are fully gapped (Fig. 1c, g, h). However, the size of the SOC gap is quite small because of the weak intrinsic SOC strength in $Cu_2Si$. After artificially increasing the SOC strength, the size of the gap increases accordingly, as shown in Fig. 1d. These results show that mirror reflection symmetry only protects the Dirac nodal loops in the absence of SOC.

To further validate the role of mirror reflection symmetry in the protection of the gapless nodal loops, we introduce artificial perturbations to break the mirror reflection symmetry. The first method involves the introduction of buckling in the honeycomb Cu lattice while keeping the Si atoms unchanged, as schematically shown in Fig. 2a. This kind of buckling in a honeycomb lattice is similar to the intrinsic buckling in silicene and germanene, with neighbouring atoms buckled upwards and downwards, respectively[29]. The second method to break the mirror reflection symmetry involves shifting the Si atoms downwards, as schematically shown in Fig. 2b. Our calculated band structures without SOC show that both nodal lines are gapped except the remaining gapless Dirac points along the Γ-M and Γ-K directions (Fig. 2c, d). These results confirm that the two gapless nodal loops in the absence of SOC are protected by the mirror reflection symmetry. The remaining gapless Dirac cones may be protected by other crystal symmetries, as discussed in the Supplementary Note 2.

**ARPES measurements on $Cu_2Si$/Cu(111).** To directly confirm the intriguing nodal loop properties in $Cu_2Si$, we performed high-resolution ARPES to measure its band structures. We synthesized monolayer $Cu_2Si$ by directly evaporating atomic Si on Cu(111) in an ultrahigh vacuum. This sample preparation method is superior to the previously reported CVD methods[27, 28], as it can avoid exotic impurities introduced by the gases. The as-prepared $Cu_2Si$ sample is of high quality (see the Supplementary Note 3 for details) and is thus suitable for the high-resolution ARPES measurements.

Monolayer $Cu_2Si$ forms a $(\sqrt{3} \times \sqrt{3})R30°$ superstructure with respect to the $Cu(111)$-$1 \times 1$ lattice, in agreement with previous reports[27, 28]. A schematic drawing of the Brillouin zones of $Cu_2Si$ and Cu(111) is shown in Fig. 3a. In Fig. 3b–e, we show the evolution of constant energy contours (CECs) with binding energies. Several pockets centred at the Γ point can be seen: one hexagon, one hexagram, and one circle. Because of the matrix element effects, the ARPES intensities are anisotropic, resulting in a rhombic shape at the Fermi level. The hexagon, hexagram and

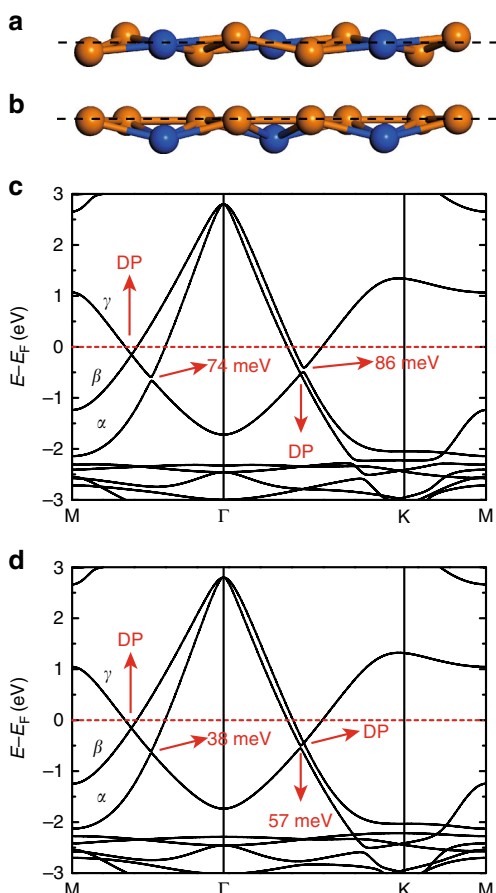

**Fig. 2** Band structures of $Cu_2Si$ without SOC after breaking the mirror reflection symmetry. **a**, **b** Two configurations that break the mirror reflection symmetry in $Cu_2Si$. **a** the neighbouring Cu atoms have a 0.1 Å buckling while the Si atoms remain in-plane; **b** the Si atoms are shifted 0.1 Å downwards. The shift of atoms is enlarged for clarity. The horizontal dashed lines indicate the original plane where all atoms were located before the shift. **c**, **d** Calculated band structures for the two configurations in **a**, **b**. The nodal lines are gapped, except for one gapless Dirac point (DP) along the Γ-M and Γ-K directions. The sizes of the gaps are indicated in **c**, **d**

circle become clearer at higher binding energies. These bands agree well with bands $\alpha$, $\beta$, and $\gamma$ from our calculations (Fig. 1e). With increasing binding energies, the sizes of the hexagon and hexagram increase, while the size of the circle decreases. In particular, the circle is larger than the hexagon and hexagram at the Fermi level and becomes smaller at binding energies higher than 1.0 eV, indicating the existence of two gapless or gapped nodal loops centred at the Γ point.

In Fig. 3f, we show the ARPES intensity plots measured along the Γ-K direction. The band crossings, i.e., the nodal lines, are clearly observed at both sides of the Γ point (indicated by the black arrows), in agreement with the evolution of the CECs from Fig. 3b–e. The band crossing is located at deeper binding energy compared with the freestanding $Cu_2Si$ (Fig. 1), which might originate from the electron doping of the metallic Cu(111) substrate. Furthermore, linear dispersion is observed near the nodal lines, in agreement with our theoretical calculation results. These results show that the quasiparticles in $Cu_2Si$ are Dirac nodal line fermions. Because of the small separation of $\alpha$ and $\beta$ bands along the Γ-K direction, the two bands are not clearly resolved in Fig. 3f. According to our theoretical calculations, the two bands are well separated along the Γ-M direction (Fig. 1). As Γ-M is a mirror axis ($M_\sigma$), one can unambiguously determine

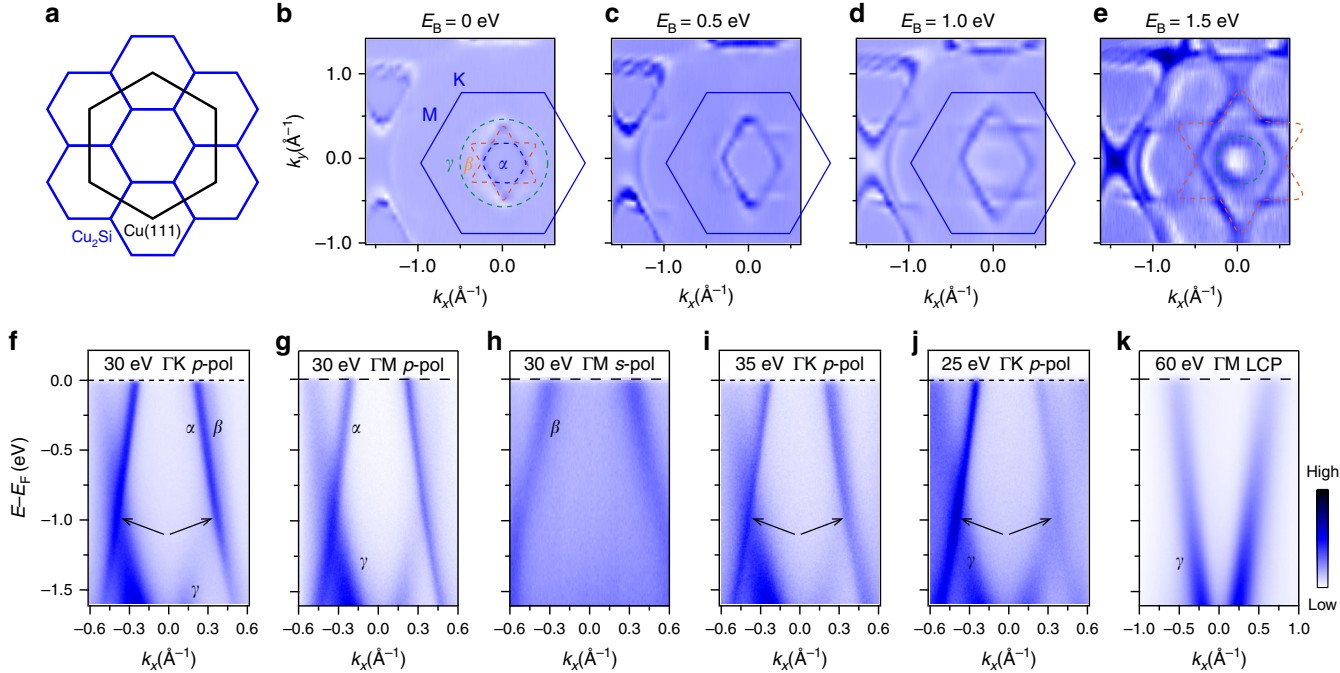

**Fig. 3** ARPES results for monolayer Cu$_2$Si on Cu(111). **a** Schematic drawing of the Brillouin zones of Cu$_2$Si (blue hexagons) and Cu(111) (black hexagon). As the lattice of Cu$_2$Si is $(\sqrt{3} \times \sqrt{3})$R30° with respect to the Cu(111)-1 × 1 lattice, the K point of Cu(111) is located at the Γ point of the second Brillouin zone of Cu$_2$Si. **b–e** Second derivative CECs measured using 30-eV $p$-polarized photons. Three closed contours have been observed: a hexagon, a hexagram, and a circle, as indicated by the dashed lines. **f, i, j** ARPES intensity plots along the Γ-K direction measured with different photon energies: 30, 35, and 25 eV, respectively. The black arrows mark the position of the crossing points. **g, h** ARPES intensity plots along the Γ-M direction measured with $p$ and $s$ polarized light, respectively. **k** ARPES intensity plots along the Γ-M direction measured with 60-eV circularly polarized light. The γ band is clearly observed while the α and β bands are suppressed

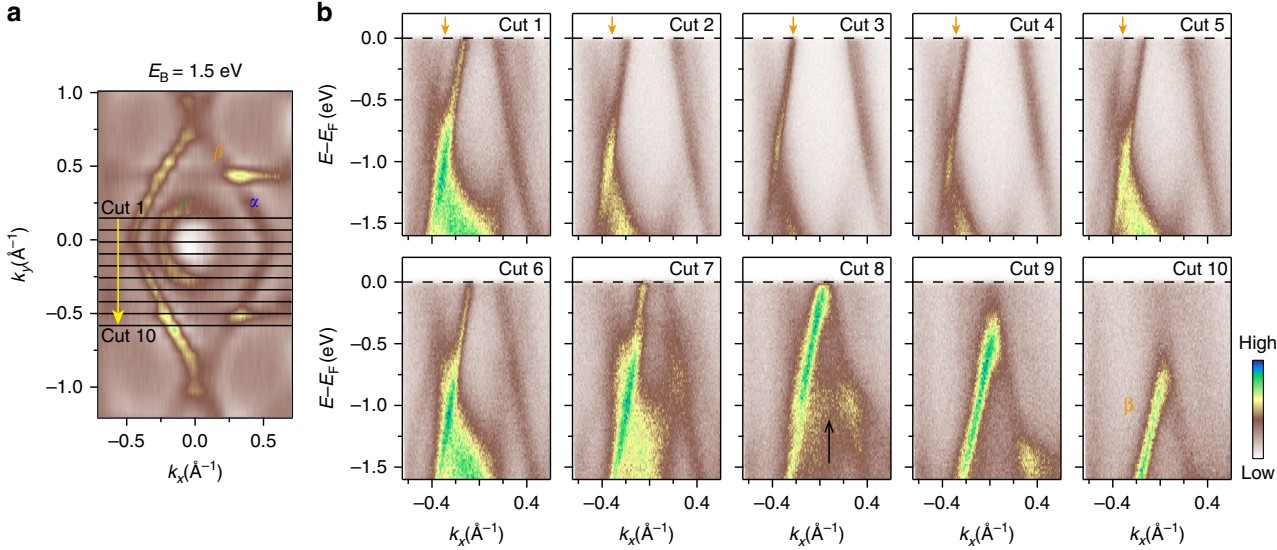

**Fig. 4** Detailed band structures of Cu$_2$Si on Cu(111). **a** Constant energy contour taken at $E_B = 1.5$ eV. The black lines (cut 1 to cut 10) indicate the positions where the ARPES intensity plots in **b** are taken. **b** ARPES intensity plots measured along cut 1 to cut 10. The orange arrows indicate the evolution of band β. The black arrow in cut 8 indicates a flat band that originates from NL1. All the data were taken with $p$-polarized light

even/odd $M_\sigma$ parity of the bands based on the polarization dependence of the matrix element. The ARPES intensity plots measured along the Γ-M direction are shown in Fig. 3g, h. We find that the α and γ bands are observed with $p$ polarized light and the β band is observed with $s$ polarized light, indicating that $M_\sigma$ parity of the α and γ bands is even, and that of the β band is odd. The result is fully consistent with our theoretical

calculations. Besides the polarization of the incident photons, the spectral weight of the bands is also dependent on the photon energy. With 60-eV photons, only the γ band is observable, while the spectral weight of the α and β bands is suppressed (Fig. 3k).

To further confirm the intriguing band structures in Cu$_2$Si, we show the ARPES intensity plots along a series of parallel momentum cuts in the Γ-K direction, as indicated in Fig. 4a.

From cut 1 to cut 10, band $\beta$ first moves closer to band $\alpha$ until $k_y = 0$ (cut 3), and then separates again. We note that cut 8 is measured along one edge of the hexagonal NL1, so one can see a relatively flat band near $E_B = 1.0$ eV (indicated by a black arrow). These results agree well with our calculated band structures and thus confirm the existence of two Dirac nodal loops in $Cu_2Si$.

## Discussion

As $Cu_2Si$ is only one atomic layer thick, all three bands are expected to have two-dimensional characteristics, i.e., no $k_z$ dispersion, which can be confirmed by ARPES measurements with different photon energies. In Fig. 3f, i, j, we present the ARPES intensity plots along the $\Gamma$-K direction measured with three different photon energies, which shows no obvious change for all the three bands. The photon energy-independent behaviour indicates no $k_z$ dispersion for the three bands, in agreement with their two-dimensional characteristics. The intensity variance of these bands originates from the matrix element effects in the photoemission process. On the other hand, the bulk bands of Cu(111) are expected to be folded to the first Brillouin zone of $Cu_2Si$ because of the Umklapp scattering of the $(\sqrt{3} \times \sqrt{3})$R30° superlattice. However, we did not observe the folded bulk bands of Cu(111) within our experimental resolution. This result indicates a weak interaction between $Cu_2Si$ and Cu(111) that results in the negligible intensity of the folded bands.

It should be noted that the mirror reflection symmetry in $Cu_2Si$ is indeed broken when it is prepared on the Cu(111) surface. This is because one side of $Cu_2Si$ is vacuum, while the other side is the Cu(111) substrate. The interaction with the substrate could open a band gap at the nodal lines because of the breaking of mirror reflection symmetry. However, we do not find a clear signature of gap opening in our ARPES measurements, which is another evidence for the weak interaction of $Cu_2Si$ and the Cu(111) substrate. Indeed, our first-principles calculation results including the substrates show that the surface $Cu_2Si$ layer remains approximately planar on Cu(111) (see Supplementary Note 4 for details), in agreement with previous theoretical and experimental results[30, 31]. These results indicate that the mirror reflection symmetry is preserved to a large extent. As a result, no obvious gap opens at the Dirac nodal lines within our experimental resolution, in agreement with our calculated band structures including the substrates (Supplementary Fig. 5). As we have discussed in Fig. 1c, SOC can also lead to gap opening at the nodal lines, but the intrinsic SOC in $Cu_2Si$ is too weak to induce detectable gaps.

Our theoretical and experimental results have unambiguously confirmed the existence of two concentric Dirac nodal loops in monolayer $Cu_2Si$. These nodal loops are protected by the mirror reflection symmetry and thus robust against symmetry conserving perturbations. These results not only extend the concept of Dirac nodal lines from three- to two-dimensional materials, but also provide a new platform to realize novel devices at the nanoscale. As copper has already been widely used for the preparation and subsequent transfer of graphene, we expect that monolayer $Cu_2Si$ will be able to be transferred to insulating substrates, which is crucial for future transport measurements and device applications. On the other hand, although the breaking of mirror reflection symmetry will destroy the global Dirac nodal loops, our theoretical calculations have found the formation of gapless Dirac cones along the $\Gamma$-M and $\Gamma$-K directions (Fig. 2). This result suggests the possibility of tuning the Dirac states in $Cu_2Si$ by controllable breaking of the mirror reflection symmetry, which might be realized by selecting appropriate substrates. Finally, we want to emphasize that the Dirac nodal lines in $Cu_2Si$ are protected by the crystalline symmetry, instead of the band topology as with generic topological semimetals. The symmetry protected Dirac nodal lines can serve as a good platform for the topological phase transition among two-dimensional Dirac nodal line, topological insulator, topological crystalline insulator and Chern insulator by inducing certain mass terms[21, 22, 32].

## Methods

**Sample preparation and ARPES measurements**. The sample preparation and photoemission measurements were performed at BL-9A (Proposal number 16AG005) and BL-9B (Proposal Number 17AG011) of HiSOR at Hiroshima University and the VUV-Photoemission beamline of the Elettra synchrotron at Trieste. Single-crystal Cu(111) was cleaned by repeated sputtering and annealing cycles. Monolayer $Cu_2Si$ was prepared by directly evaporating Si on Cu(111) while keeping the substrate at approximately 500 K. The structure of $Cu_2Si$ was confirmed by the appearance of sharp $(\sqrt{3} \times \sqrt{3})$R30° low-energy electron diffraction (LEED) patterns. The pressure during growth was less than $1 \times 10^{-7}$ Pa. After preparation, the sample was directly transferred to the ARPES chamber without breaking the vacuum. During the ARPES measurements, the sample was kept at 40 K. The energy resolution was better than 20 meV. The pressure during measurements was below $2 \times 10^{-9}$ Pa.

**First-principles calculations**. Our first-principles calculations were carried out using VASP (Vienna ab-initio simulation package)[33] within the generalized-gradient approximation of the Perdew, Burke, and Ernzerhof[34] exchange-correlation potential. A cutoff energy of 450 eV and a k-mesh of $27 \times 27 \times 1$ were chosen for self-consistent-field calculations. The lattice constant of 4.123 Å was obtained from the experimental value, and the thickness of vacuum was set to 18 Å, which is adequate to simulate two-dimensional materials. The convergence criteria of total energy and the force of each atom were 0.001 eV and 0.01 eV Å$^{-1}$, respectively. The spin-orbital effect was considered in part of our calculations.

**Data availability**. The data sets generated during and/or analyzed during the current study are available from the corresponding authors on reasonable request.

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

## Acknowledgements

We thank Professor X. J. Zhou for providing the Igor macro to process the ARPES data. This work was supported by the Ministry of Education, Culture, Sports, Science and Technology of Japan (Photon and Quantum Basic Research Coordinated Development Program), the MOST of China (Grant Nos. 2014CB920903, 2016YFA0300600, 2013CBA01601, and 2016YFA0202300), the NSF of China (Grant Nos. 11574029, 11225418, 11674366, and 11674368), the Strategic Priority Research Program of the Chinese Academy of Sciences (Grant No. XDB07020100), and the US Department of Energy (DOE), Office of Science (OS), Office of Basic Energy Sciences, Division of Materials Science and Engineering (Grant No. DE-FG02-07ER46383).

## Author contributions

B. Feng, L.C. and K.W. conceived the research. B. Feng, S.I., Y.F., S.W., S.K.M., P.S., P.M., M.A. and I.M. performed the ARPES measurements. B.Fu, S.K., C.-C.L., O.S. and Y.Y. performed the theoretical calculations. P.C., K.W. and L.C. performed the STM experiments. All authors contributed to the discussion of the data. B. Feng wrote the manuscript with contributions from all authors.

## Additional information

**Competing interests:** The authors declare no competing financial interests.

