## [Peer Review File · Nature Communications]

Reviewers' comments:

Reviewer #1 (Remarks to the Author):

In the paper, the authors present an important experimental observation that supports the existence of one-dimensional Dirac line node semimetals in two dimension. They identify a single layer of Cu₂Si can host Dirac line nodes in its two-dimensional momentum space, protected by a mirror symmetry. Considering the current interest in topological semimetals including Dirac line node semimetals and the novelty of their finding, I would like to recommend to publish the paper in Nature Communication if the authors properly address the following criticism.

1. In the introduction, authors cite topological semimetals, but they missed important examples, such as Cu₃NPd. Author should cite the following references:

- i. Physical Review Letters,
- ii. Physical Review Letters,

2. The Dirac semimetal phase that authors present can be understood as a symmetry-protected semimetal that is distinct from generic topological semimetals, in the sense that the Dirac line nodes in Cu₂Si is not protected by band topology, but protected by only crystalline symmetry. Whereas the symmetry-protected Dirac line node is still an important class of semimetals, exhibiting protected bulk metallicity, their distinction should be presented clearly. Therefore, I suggest to include the discussion clarifying this point in the main text.

3. Readers should be able to better understand the material if the authors present the orbital characters of the conduction and valance bands of Cu₂Si. I would like to suggest to present the orbital-projected band structure with different color schemes, which should clarify the orbital characters of the conduction and valence bands.

4. It appears that p orbitals (p_x, p_y, p_z) of Si atoms that form the triangular lattice should describe the low-energy electronic band structure. Based on the above analysis, I'd like to suggest to develop a tight-binding model and low-energy k-p theory that describe the Dirac line nodes?.

5. Authors show that, by losing the mirror symmetry, the Dirac line nodes become gapped except some points, thus forming Dirac points. What is the origin of the protection of these Dirac points? Are they protected by any crystalline symmetries that the authors imposed when introducing mirror-breaking perturbation? It appears that inversion and time-reversal symmetry play a crucial role to protect the Dirac points, similar to graphene. Can authors develops a similar argument about the topological or symmetrical origin of the Dirac points appearing in the absence of mirror symmetry?

6. Author claims that the presence of Cu substrate gives rise to a negligible effect on the presence of Dirac line nodes in Cu₂Si. Can authors demonstrate this quantitatively by using Density functional theory supercell calculations including both the substrate and Cu₂Si?

In summary, I would like to suggest a minor revision of the manuscript, properly responding to the above criticism.

Reviewer #2 (Remarks to the Author):

The manuscript reports band structure calculations and ARPES measurements on Cu₂Si monolayer film grown on Cu substrate. The calculation results reveal three types of topologically non-trivial bands (i.e. alpha, beta and gamma bands) and these bands show linear crossing near the Fermi level. These Dirac band crossing leads to two Dirac nodal lines, which are protected by mirror symmetry. These predicted results look interesting. However, their predicted results are not quite supported by their ARPES experimental data.

1. The ARPES data presented in this manuscript have low quality. The hexagon, hexagram and circle Fermi surface contours claimed by the authors are barely discernable in Fig. 3b-3e. The Dirac band crossing points are hardly seen in Fig. 3f-3i due to the low quality of the data.

2. Although the calculation clearly shows alpha and beta bands are markedly separated (Fig. 1b and 1c), this is not clearly resolved in the experimental data shown in Fig. 3f, 3g or 3j. Though

Fig.4 shows traces suggesting the evolution of beta band, again, the data quality is well low, which does not provide clear support for their claim.

3. The SOC-induced gap due to the breaking mirror symmetry is not observed in the experimental data. The authors interpreted it as weak coupling between the Cu₂Si monolayer and the Cu substrate. This interpretation does not look convincing.

Unlike other established 2D (graphene)/3D Dirac materials whose Dirac fermion properties such as light effective mass and high mobility can be confirmed with magnetotransport measurements, the Cu₂Si monolayer material reported in this manuscript does not allow for transport verification for its Dirac fermion properties. This is because that the film is grown on Cu substrate which is conducting, which prevents transport property characterization for the film. Though the authors said they expect to transfer the monolayer film to an insulating substrate, I suspect this is feasible.

Given the reasons I summarized above, I do not feel this work could impact the field. It might deserve publication in a more specialized journal, but it does not meet high standard of Nature Communications. Therefore, I do not recommend this manuscript for publication in Nature Communications.

Reviewer #3 (Remarks to the Author):

Feng et al. reported the theoretical calculations and the experimental realization of a two-dimensional Dirac nodal line semimetal Cu₂Si, which is the 2D counterpart of bulk Dirac nodal line semimetals. This study is of scientific interests, however, the current article has a few flaws, as commented below.

1. As shown in the theoretical calculations, the nodal lines are gapped due to the SOC. The authors also stated in their discussion that "SOC can lead to gap opening at the nodal lines, but the intrinsic SOC in Cu₂Si is too weak to induce detectable gaps". In this sense, the hypothetical Cu₂Si without SOC is a 2D topological nodal line semimetal, but the real Cu₂Si material (with intrinsic SOC) is not. Therefore I think the importance of this work becomes somehow irrelevant, since the foundation of the experimental part is not solid.

2. For fig.3f, the authors stated that the band crossing are clearly observed and the linear dispersion is observed near the nodal lines. I do not find these statements are obvious due to the very limited data analysis presented. i.e. the gamma band is very broad and the spectral weight seems to be even suppressed near the supposed crossing, therefore I do not see a "clear" band crossing.

3. In line 116 as described for fig.4, "cut 8 is measured along one edge of the hexagonal NL1". If I'm not mistaken, cut 8 is along the edge of NL2 (the blue hexagon) as defined by the authors. Furthermore, the orange arrows indicate the evolution of the beta band, where I suppose the band with the most prominent feature from cut 1-7 should be the alpha band. In cut 8, the flat band indicated by the black arrow is originated from NL2, which is supposed to be from alpha. Therefore it is not that easy to understand the evolution of alpha and beta from cut 7 to 8, based on the presented explanations.

4. Although I am convinced that there are two bands alpha and beta close to each other along the GammaK direction, the markers indicating the beta band in fig.3j do not appear to be straightforward.

Therefore, from my point of view, I do not recommend its publication in Nature Communications in the current form.

Reply to Referee 1:

C1: *In the paper, the authors present an important experimental observation that supports the existence of one-dimensional Dirac line node semimetals in two dimension. They identify a single layer of Cu₂Si can host Dirac line nodes in its two-dimensional momentum space, protected by a mirror symmetry. Considering the current interest in topological semimetals including Dirac line node semimetals and the novelty of their finding, I would like to recommend to publish the paper in Nature Communication if the authors properly address the following criticism.*

Reply: Thanks for the careful review of our manuscript. We will address all the criticism in the following.

C2: *In the introduction, authors cite topological semimetals, but they missed important examples, such as Cu₃NPd. Author should cite the following references:*

>> i. Physical Review Letters,

>> ii. Physical Review Letters,

Reply: Thanks for the suggestion. We have cited the two papers in our revised manuscript.

C3: *The Dirac semimetal phase that authors present can be understood as a symmetry-protected semimetal that is distinct from generic topological semimetals, in the sense that the Dirac line nodes in Cu₂Si is not protected by band topology, but protected by only crystalline symmetry. Whereas the symmetry-protected Dirac line node is still an important class of semimetals, exhibiting protected bulk metallicity, their distinction should be presented clearly. Therefore, I suggest to include the discussion clarifying this point in the main text.*

Reply: Thanks for the valuable suggestion. We have revised our manuscript accordingly:

Line 149: we changed the sentences: “*For the topological...certain mass terms.*” to “*Finally, we want to emphasize that the Dirac nodal lines in Cu₂Si are protected by the crystalline symmetry, instead of the band topology as with generic topological semimetals. The symmetry protected Dirac nodal lines can serve as a good platform for the topological phase transition among two-dimensional Dirac nodal line, topological insulator, topological crystalline insulator and Chern insulator by inducing certain mass terms.*”

We hope that our manuscript can help the readers to better understand the protection mechanism in the related materials.

C4: *Readers should be able to better understand the material if the authors present the orbital characters of the conduction and valance bands of Cu₂Si. I would like to suggest to present the orbital-projected band structure with different color schemes, which should clarify the orbital characters of the conduction and valence bands.*

Reply: We agree with the referee that the orbital character of the bands is important. So we followed the referee’s suggestion, and calculated the orbital projected band structures, as shown in Fig. R1. The M_z parity of the orbitals further supports our conclusion on the M_z parity of the three bands. The related discussion has been added in the third section of the supplementary information.

Fig. R1: Orbital projected band structures of free-standing Cu_2Si . Left: Si orbitals; Right: Cu orbitals.

C5: It appears that p orbitals (p_x , p_y , p_z) of Si atoms that form the triangular lattice should describe the low-energy electronic band structure. Based on the above analysis, I'd like to suggest to develop a tight-binding model and low-energy k - p theory that describe the Dirac line nodes?.

Reply: Thanks for the suggestion. According to the referee's suggestion, here we develop a simple tight-binding model based on the p orbitals of Si. The basis we use is $[p_z, p_x, p_y]$. Considering the nearest neighbour coupling, the Hamiltonian can be written as:

$$H_0 = \begin{bmatrix} h_{11} & h_{12} & h_{13} \\ h_{12}^* & h_{22} & h_{23} \\ h_{13}^* & h_{23}^* & h_{33} \end{bmatrix}$$

$$h_{11} = \varepsilon_1 + t[\exp(-ikR_1) + \exp(-ikR_2) + \exp(-ikR_3) + \exp(-ikR_4) + \exp(-ikR_5) + \exp(-ikR_6)]$$

$$h_{22} = \varepsilon_2 + t_\sigma[\exp(-ikR_1) + \exp(-ikR_4)] + (0.75t_\pi + 0.25t_\sigma)[\exp(-ikR_2) + \exp(-ikR_5) + \exp(-ikR_3) + \exp(-ikR_6)]$$

$$h_{33} = \varepsilon_3 + t_\pi[\exp(-ikR_1) + \exp(-ikR_4)] + (0.25t_\pi + 0.75t_\sigma)[\exp(-ikR_2) + \exp(-ikR_5) + \exp(-ikR_3) + \exp(-ikR_6)]$$

$$h_{23} = \frac{\sqrt{3}}{4}(-t_\pi + t_\sigma)[\exp(-ikR_2) + \exp(-ikR_5)] + \frac{\sqrt{3}}{4}(t_\pi - t_\sigma)[\exp(-ikR_3) + \exp(-ikR_6)]$$

$$h_{12} = 0$$

$$h_{13} = 0$$

where $\varepsilon_{1,2,3}$ are the onsite energies of basis; t represents the isotropic coupling between the p_z orbitals; t_σ and t_π represent the σ bond and the π bond of in plane p_x/p_y orbitals, respectively. Due to the opposite M_z symmetry of p_z and p_x/p_y basis, $h_{12} = h_{13} = 0$. Those zero coupling is responsible for the existence of gapless nodal lines. With proper onsite energy and hopping parameters, the γ band will cross the α and β bands without opening a gap, forming the two Dirac nodal lines [Fig. R2(b)]. Besides, by introducing nonzero h_{12}, h_{13} term, the nodal lines will be gapped except for certain k -points, as shown in Fig. R2(c).

It should be noted that these analyses are only qualitatively consistent with our DFT calculations, the strong anisotropy of the α and β bands and the hybridization around the K points in DFT has not been reproduced. An accurate tight-binding model needs participation of Cu orbitals, which is too complex to adopt the model, and the result should also reproduce our DFT results. Considering only the Si p orbitals, as suggested by the referee, the tight-binding model already gives enough accuracy for describing the low-energy band structures of Cu_2Si , which is consistent with the orbital analysis in the supplementary information.

Fig. R2: (a) Black rhombus: the Brillouin zone of Cu_2Si , which contains one Si atom. a corresponds to the lattice constant. R_1 - R_6 represent the six neighbouring Si atoms. (b) Tight-binding band structures considering the Si p orbitals only. The parameters in the calculations are $\epsilon_1=0.3$; $\epsilon_2=\epsilon_3=-0.65$; $t=-0.36$; $t_\sigma=0.72$, $t_\pi=0.44$. (c) Tight-binding band structures of Cu_2Si when $h_{12} \neq 0$; $h_{13} \neq 0$. But the system has C_6 symmetry.

C6: Authors show that, by losing the mirror symmetry, the Dirac line nodes become gapped except some points, thus forming Dirac points. What is the origin of the protection of these Dirac points? Are they protected by any crystalline symmetries that the authors imposed when introducing mirror-breaking perturbation? It appears that inversion and time-reversal symmetry play a crucial role to protect the Dirac points, similar to graphene. Can authors develop a similar argument about the topological or symmetrical origin of the Dirac points appearing in the absence of mirror symmetry?

Reply: Thanks for the suggestion. We agree that the remaining gapless Dirac points are interesting and deserve further investigation. Our extensive analysis shows that these Dirac points are protected by out-of-plane mirror symmetry or C_2 rotation symmetry. The detailed discussion has been added in the fifth part of the supplementary information.

C7: Author claims that the presence of Cu substrate gives rise to a negligible effect on the presence of Dirac line nodes in Cu_2Si . Can authors demonstrate this quantitatively by using Density functional theory supercell calculations including both the substrate and Cu_2Si ?

Reply: Thanks for the suggestion. We have performed calculations including the substrate, and the new results have been added in the fourth part of the supplementary information. The calculated band structures are in excellent agreement with our ARPES data. Our calculations also indicate a relatively weak coupling between the Cu_2Si monolayer and the substrate, which is the reason for the survival of the Cu_2Si bands on $\text{Cu}(111)$.

Reply to Referee 2:

C1: The manuscript reports band structure calculations and ARPES measurements on Cu_2Si monolayer film grown on Cu substrate. The calculation results reveal three types of topologically non-trivial bands (i.e. alpha, beta and gamma bands) and these bands show linear crossing near the Fermi level. These Dirac band crossing leads to two Dirac nodal lines, which are protected by mirror symmetry. These predicted results look interesting. However, their predicted results are not

quite supported by their ARPES experimental data.

Reply: Thanks for the careful review of our manuscript. We will address all the criticism in the following.

C2: *The ARPES data presented in this manuscript have low quality. The hexagon, hexagram and circle Fermi surface contours claimed by the authors are barely discernable in Fig. 3b-3e. The Dirac band crossing points are hardly seen in Fig. 3f-3i due to the low quality of the data.*

Reply: We agree that the quality of our raw ARPES data in the previous version is not very high. To present the data more clearly, we changed Figs. 3b-3e to the second derivative CECs. In the new figures, the hexagon (α band), hexagram (β band) and circle (γ band) are clearly resolved. For example, we show the CEC at $E_B=1.5$ eV in Fig. R3.

We also show the second derivative ARPES intensity plots in Figs. 3g and 3j. In these new figures, the crossing of the bands is clearly resolved.

Fig. R3: CEC at $E_B=1.5$ eV. (a) without guidelines. (b) with guidelines.

C3: *Although the calculation clearly shows alpha and beta bands are markedly separated (Fig. 1b and 1c), this is not clearly resolved in the experimental data shown in Fig. 3f, 3g or 3j. Though Fig.4 shows traces suggesting the evolution of beta band, again, the data quality is well low, which does not provide clear support for their claim.*

Reply: Our calculation shows that α and β bands are markedly separated only in the ΓM direction. In the ΓK direction, the two bands are very close to each other, which is beyond the resolution our ARPES measurements. In the ΓM direction (Fig. 3h in the new version), our ARPES data clearly show the separation of the two bands. So our data agree well with the calculation results.

The data in Fig. 4 have shown the evolution of the β band. In Fig. R4, we show the second derivative ARPES intensity plot for cut 1 in Fig. 4. The two bands are clearly resolved, as indicated by the red and blue dashed lines in Fig. R4(c).

Fig. R4: (a) The same ARPES intensity plots as cut 1 in Fig. 4. (b) the second derivative image. (c) the same with (b) but with red and blue dashed line to indicate the alpha and beta bands, respectively.

C4: *The SOC-induced gap due to the breaking mirror symmetry is not observed in the experimental data. The authors interpreted it as weak coupling between the Cu₂Si monolayer and the Cu substrate. This interpretation does not look convincing.*

Reply: We agree with the referee that we did not provide convincing arguments on the negligible gap. In the revised manuscript, we have calculated the band structures including the substrate and the new results have been included in the supplementary information [Fig. S4]. After structure optimization, we found that the Cu₂Si layer remains relatively flat with only small buckling ($<0.07\text{\AA}$), which means that the mirror symmetry is approximately preserved. Moreover, our calculated band structures including the substrate show negligible gap opening at the nodal lines, in agreement with our experimental results. These new calculation results suggest a weak coupling between the Cu₂Si monolayer and the Cu substrate and can excellently explain the negligible gap opening in our ARPES data.

Finally, we want to emphasize that the SOC-induced gap is not due to the breaking of mirror symmetry (M_z). The SOC or breaking of M_z could independently induce a gap. Due to the intrinsically small SOC strength, the SOC-induced gap is too small to be detected. What the referee mentioned, we think, is the gap induced by the breaking of M_z . We have added new calculations to support our conclusion, as discussed above.

C5: *Unlike other established 2D (graphene)/3D Dirac materials whose Dirac fermion properties such as light effective mass and high mobility can be confirmed with magnetotransport measurements, the Cu₂Si monolayer material reported in this manuscript does not allow for transport verification for its Dirac fermion properties. This is because that the film is grown on Cu substrate which is conducting, which prevents transport property characterization for the film. Though the authors said they expect to transfer the monolayer film to an insulating substrate, I suspect this is feasible.*

Given the reasons I summarized above, I do not feel this work could impact the field. It might deserve publication in a more specialized journal, but it does meet high standard of Nature Communications. Therefore, I do not recommend this manuscript for publication in Nature Communications.

Reply: We agree that monolayer Cu₂Si prepared on Cu(111) does not allow for transport

measurements. However, we believe that our results are still important enough:

1. Our results confirm the first two-dimensional material that host nodal lines, which is of fundamental importance. The transport measurements call for further experimental efforts, which is out of the scope of the present study.
2. Although the Cu(111) substrate do not allow for transport measurements, it is an ideal substrate for our ARPES and STM measurements because insulating substrates will have charging effects. It should be noted that many novel materials are first prepared on conducting substrates, such as artificial graphene [Nature 483, 306 (2012)], borophene [Science 350, 1513 (2015)] and stanene [Nat. Mater. 14, 1020 (2015)]. The STM or ARPES characterization of these novel materials have already stimulated great scientific interest. We believe that our discovery of 2D nodal lines will also highly impact this field.
3. Indeed, monolayer Cu_2Si can also be synthesized on semiconductor Si(111), which have already been introduced in our manuscript. The semiconductor Si(111) allows for transport measurements.

Reply to Referee 3:

C1: *Feng et al. reported the theoretical calculations and the experimental realization of a two-dimensional Dirac nodal line semimetal Cu_2Si , which is the 2D counterpart of bulk Dirac nodal line semimetals. This study is of scientific interests, however, the current article has a few flaws, as commented below.*

Reply: Thanks for the careful review of our manuscript. We will address all the criticism in the following

C2: *As shown in the theoretical calculations, the nodal lines are gapped due to the SOC. The authors also stated in their discussion that “SOC can lead to gap opening at the nodal lines, but the intrinsic SOC in Cu_2Si is too weak to induce detectable gaps ” . In this sense, the hypothetical Cu_2Si without SOC is a 2D topological nodal line semimetal, but the real Cu_2Si material (with intrinsic SOC) is not. Therefore I think the importance of this work becomes somehow irrelevant, since the foundation of the experimental part is not solid.*

Reply: We agree that the nodal lines in Cu_2Si will be gapped in the presence of SOC. However, the SOC induced gap is very small because of the intrinsically small SOC strength in Cu_2Si . So Cu_2Si can be *approximately* considered to host gapless nodal lines, and these approximated nodal lines have been directly confirmed by our ARPES measurements.

Similarly, graphene also has a gap at the Dirac point if SOC is taken into account. But many of the interesting properties of graphene are directly related to the massless Dirac fermions, while the gap effects are negligible. We also realized that many 3D nodal line semimetals are only valid in the absence of SOC, such as fcc alkaline earth metals [Nat. Commun. 8, 14022 (2017)], ZrSiS [Nat. Commun. 7, 11696 (2016)], Cu_3PdN [Phys. Rev. Lett. 115, 036806 (2015); Phys. Rev. Lett. 115, 036807 (2015)], and so on. Despite the existence of the SOC-induced gap, transport measurements have revealed intriguing properties that are related to the nodal lines [Sci. Adv. 2, 1601742 (2016); Phys. Rev. Lett. 117, 016602 (2016)], and the gap effects could also be neglected.

So we believe that the “gapless” Dirac nodal lines in Cu₂Si we observed is important, which may determine its intriguing transport properties.

C3: For fig.3f, the authors stated that the band crossing are clearly observed and the linear dispersion is observed near the nodal lines. I do not find these statements are obvious due to the very limited data analysis presented. i.e. the gamma band is very broad and the spectral weight seems to be even suppressed near the supposed crossing, therefore I do not see a “clear” band crossing.

Reply: We agree that Fig. 3f is not very clear. To show the crossing more clearly, we present the second derivative images measured with 30-eV and 35-eV photons in Fig. R5. Now the band crossing is clearly resolved. We have added these figures in the revised manuscript [Figs. 3g and 3j].

Fig. R5: Second derivative ARPES intensity plots along the Γ K direction measured with 30-eV (left) and 35 eV (right) photons.

C4: In line 116 as described for fig.4, “cut 8 is measured along one edge of the hexagonal NL1”. If I’m not mistaken, cut 8 is along the edge of NL2 (the blue hexagon) as defined by the authors.

Reply: Thanks for pointing out our mistake. In the revised manuscript, we switched NL1 and NL2 in Fig. 1(f). This change is also in agreement with the description in Line 64.

C5: Furthermore, the orange arrows indicate the evolution of the beta band, where I suppose the band with the most prominent feature from cut 1-7 should be the alpha band. In cut 8, the flat band indicated by the black arrow is originated from NL2, which is supposed to be from alpha. Therefore it is not that easy to understand the evolution of alpha and beta from cut 7 to 8, based on the presented explanations.

Reply: From cut 7 to 8, it is near NL1 (corrected in the revised version). The β band is also relatively flat because the cut is parallel to the edges of the hexagram. So the bands change dramatically around cuts 7 and 8. From cut 1 to 7, the most prominent band is α . From cut 8, the most prominent band is suddenly changed to β . This change is not obvious because the edges of both the hexagon (α) and hexagram (β) are parallel to our measurement direction.

C6: Although I am convinced that there are two bands alpha and beta close to each other along the Gamma K direction, the markers indicating the beta band in fig.3j do not appear to be straightforward.

Reply: We agree with the referee that the markers in Fig. 3j (previous version) are not straightforward. This is because the separation of the two bands is too small. So we deleted this figure because the existence of the two bands has already been clearly resolved by the measurements along other cuts.

Summary of changes:

1. We added additional theoretical calculations in the supplementary information.
2. We switched NL1 and NL2 in Fig. 1f.
3. We changed Figs. 3b-3e to the second derivative CECs.
4. We added the second derivative band structures for Figs. 3f and 3i (the revised version).
5. We deleted the momentum distribution curves in Fig. 3.
6. Line 18: We changed the sentence: “*However, ...is still lacking.*” to “*However, in two-dimensional materials, experimental research on nodal line fermions is still lacking.*”.
7. Line 27: we deleted the words: “or crystallographic symmetries”.
8. Line 110: we deleted the sentence: “*However, from the momentum distribution...bands.*”
9. Line 121: we changed the sentence “*In Figs. 3(f)...photon energies.*” to “*In Figs. 3(f), 3(i) and 3(k), we present the ARPES intensity plots along the Γ -K direction measured with three different photon energies, which shows no obvious change for all the three bands.*”
10. Line 123: we deleted the sentence: “From the MDC...Fig. 3(i).”
11. Line 124: we changed the sentences: “*Indeed, previous...experimental resolution.*” to “*Indeed, our first-principles calculation results including the substrates show that the surface Cu_2Si layer remains approximately planar on $\text{Cu}(111)$ (see Supplementary Information for details), in agreement with previous theoretical and experimental results [29, 30]. These results indicate that the mirror reflection symmetry is preserved to a large extent. As a result, no obvious gap opens at the Dirac nodal lines within our experimental resolution, in agreement with our calculated band structures including the substrates (Fig. S4).*”
12. Line 149: we changed the sentences: “*For the topological...certain mass terms.*” to “*Finally, we want to emphasize that the Dirac nodal lines in Cu_2Si are protected by the crystalline symmetry, instead of the band topology as with generic topological semimetals. The symmetry protected Dirac nodal lines can serve as a good platform for the topological phase transition among two-dimensional Dirac nodal line, topological insulator, topological crystalline insulator and Chern insulator by inducing certain mass terms.*”
13. We have cited the two references as suggested by Referee 1.
14. We added a footnote to declare that Baojie Feng and Botao Fu as equal contribution authors.

Reviewers' comments:

Reviewer #1 (Remarks to the Author):

I think the authors responded well to the criticism I raised. I would like to suggest to accept the paper for publication.

Reviewer #2 (Remarks to the Author):

In my first report, I indicated the main result reported in this manuscript, i.e. the prediction of 2D nodal line state in the Cu₂Si monolayer is interesting. But this predicted result is not strongly supported by the ARPES data due to the low-quality of the data. In particular, both the predicted hexagon/hexagram Fermi surfaces and Dirac band crossings were not clearly resolved.

In the revised manuscript, the authors attempted improving the data quality by taking second derivative ARPES intensity plots. This approach helped resolve those predicted features to some extent. The predicted hexagon and hexagram Fermi surfaces can now be better resolved in the second derivative constant energy contours. However, those predicted Dirac band crossing along Γ -K and Γ -M are still not clearly resolved even in the second derivative plots. For instance, although the theoretical calculations predicted two crossing points along the Γ -M direction, one slightly below the Fermi level and the other near -0.8V, the ARPES experiment showed only one crossing point near -1eV. The authors also indicated that it is electron doping by the substrate that makes the Dirac node be further away from the Fermi level. In this case, the Dirac node expected to be near the Fermi level should also be visible. The other problem is that Γ -band in Fig. 3g and 3j is not clear at all. All of these problems are due to the low-quality of the data, which the authors agree with. Compared to the ARPES papers I read in high impact journals, I feel the quality of the ARPES data reported in this manuscript does not reach the high standard of high impact journals like Nature Communications. I suggest the authors to perform further experiments to improve the data quality so that they can make more quantitative comparison with their theoretical results. When the authors reach that point, I would recommend it for publication in Nature communications.

Reviewer #3 (Remarks to the Author):

The revised manuscript presents better data analysis and appearance for the readers. The points raised previously are also satisfactorily addressed.

Considering the novelty and impact of this work, I recommend it to be published on Nature Communications.

Reply to Referee 2:

Comment: *In my first report, I indicated the main result reported in this manuscript, i.e. the prediction of 2D nodal line state in the Cu₂Si monolayer is interesting. But this predicted result is not strongly supported by the ARPES data due to the low-quality of the data. In particular, both the predicted hexagon/hexagram Fermi surfaces and Dirac band crossings were not clearly resolved.*

In the revised manuscript, the authors attempted improving the data quality by taking second derivative ARPES intensity plots. This approach helped resolve those predicted features to some extent. The predicted hexagon and hexagram Fermi surfaces can now be better resolved in the second derivative constant energy contours. However, those predicted Dirac band crossing along Gama-K and Gama-M are still not clearly resolved even in the second derivative plots. For instance, although the theoretical calculations predicted two crossing points along the Gama-M direction, one slightly below the Fermi level and the other near -0.8V, the ARPES experiment showed only one crossing point near -1eV. The authors also indicated that it is electron doping by the substrate that makes the Dirac node be further away from the Fermi level. In this case, the Dirac node expected to be near the Fermi level should also be visible. The other problem is that Gama-band in Fig. 3g and 3j is not clear at all. All of these problems are due to the low-quality of the data, which the authors agree with. Compared to the ARPES papers I read in high impact journals, I feel the quality of the ARPES data reported in this manuscript does not reach the high standard of high impact journals like Nature Communications. I suggest the authors to perform further experiments to improve the data quality so that they can make more quantitative comparison with their theoretical results. When the authors reach that point, I would recommend it for publication in Nature communications.

Reply: Thanks for the comments on the quality of the data. We followed the referee's suggestion, and performed further experiments in BL-9B of Hiroshima Synchrotron Radiation Center. Our new results show that the main reason why we did not clearly observe the β band along the ΓM direction is the matrix element effect in the photoemission process. In the new experiments, we used different polarizations (linearly/circularly) at various photon energies, and all the three bands can be clearly resolved with appropriate photons.

Indeed, all the data presented in the previous version were taken with p -polarized light. As the ΓM axis is a mirror axis, each band along the ΓM direction have defined parity. The bands with even and odd parity can be observed using p and s polarized light, respectively. Using p polarized light, we can resolve the α and γ bands, while the β band is missing; using s polarized light, we can resolve the β band, while the α and γ bands are missing, as shown in Figs. R1(a) and R1(b). These results indicate that the parity of α and γ bands is even and that of β band is odd, in agreement with our theoretical calculations. When we avoid the mirror plane, the even and odd parity will mix, so we can observe all the bands with p (or s) polarized light. This is the reason why we can observe all the bands in Fig. 3(h) in the previous version, although the β band is not clear enough at that time.

To observe the three bands simultaneously along the ΓM direction, we used circularly polarized light because we can detect both even and odd parity bands. As shown in Figs. R1(c) and R1(d), all the three bands can be clearly observed. The three bands are better resolved in Figs. R2(a) and R2(b), which was measured along a different azimuth angle. The red arrows in Fig. R2(b) indicate the two crossings.

The referee also mentioned that the γ band was not clear enough. In our experiments, we found that the intensity of the γ band is stronger using higher photon energy, which is also due to the matrix element effect. In Fig. R2(c), we show the band structures measured with a photon energy of 60 eV. The γ band is clearly resolved, while the spectral weight of the α and β bands vanishes. In Fig. R2(d), we show the fitted E- k dispersion along the Γ M direction from the momentum distribution curves for the three bands, which agrees well with our theoretical calculations.

We thank the referee for suggesting further experiments, which provides solid evidence to support our conclusions. In the revised manuscript, we have changed Fig. 3 with new experimental data and added discussions on the parity of the bands in the main text.

Fig. R1: ARPES intensity plots measured with 30-eV photons. (a-d) Γ M direction; (e-h) Γ K direction. (a,e) p -polarized light. (b,f) s -polarized light. (c,g) Left circularly polarized light. (d,h) Right circularly polarized light.

Fig. R2: (a) ARPES intensity plots along the red line in the inset. (b) MDC second derivative image of (a). The red arrows indicate the crossing points. (c) ARPES intensity plots measured along the Γ M direction with circularly polarized light. (d) Fitted E- k dispersion for the three bands from the peaks in the momentum distribution curves. The α band was fitted from the right side of Fig. R1(a); the β band was fitted from the right side of Fig. R1(b); the γ band was fitted from the right side of Fig. R2(c).

Summary of changes:

1. We added five coauthors who participated in the new experiments and provided helpful discussions: Dr. Ya Feng, Mr. Shilong Wu, Prof. Kenya Shimada, Prof. Koji Miyamoto and Prof. Taichi Okuda. We also added their contributions in the “Author contributions” section.
2. We added an affiliation for author Baojie Feng.
3. We changed Fig. 3 with new data. The figure caption and has been changed accordingly.

4. Line 113: after “*the two bands are not clearly resolved in Fig. 3(f).*”, we added sentences:
“According to our theoretical calculations, the two bands are well separated along the Γ -M direction (Fig. 1). As Γ -M is a mirror axis (M_σ), one can unambiguously determine even/odd parity of the bands based on the polarization dependence of the matrix element. The ARPES intensity plots measured along the Γ -M direction are shown in Figs. 3(g) and 3(h). We find that the α and γ bands are observed with p polarized light and the β band is observed with s polarized light, indicating that the α and γ bands are even, and the β band is odd under the mirror operation with respect to the mirror axis. The result is fully consistent with our theoretical calculations. Besides the polarization of the incident photons, the spectral weight of the bands is also dependent on the photon energy. With 60-eV photons, only the γ band is observable, while the spectral weight of the α and β bands is suppressed [Fig. 3(k)].”
5. We add a sentence at the end of the figure caption of Fig. 4: “*All the data were taken with p -polarized light.*”
6. Line 163: we added “(Proposal number 16AG005) and BL-9B (17AG011)” after “BL-9A”.
7. In the caption of Fig. 2, we changed “with and without” to “without and with”.

REVIEWERS' COMMENTS:

Reviewer #2 (Remarks to the Author):

The revised manuscript has been significantly improved. In particular, the new experiments performed by the authors have clearly resolved all the three predicted bands. The new data in Fig. 3 are of good quality and strongly support their claim, i.e. experimental realization of 2D Dirac nodal line in Cu₂Si. I can now recommend this manuscript for publication in Nature Communications.

One suggestion: the compound name Cu₂Si should be included in the title.

Response to Reviewer #2

Comments: *The revised manuscript has been significantly improved. In particular, the new experiments performed by the authors have clearly resolved all the three predicted bands. The new data in Fig. 3 are of good quality and strongly support their claim, i.e. experimental realization of 2D Dirac nodal line in Cu₂Si. I can now recommend this manuscript for publication in Nature Communications.*

One suggestion: the compound name Cu₂Si should be included in the title.

Reply: We thank the referee for the recommendation. We followed his/her advice and changed the title to “Experimental realization of two-dimensional Dirac nodal line fermions in monolayer Cu₂Si”.